# Lexicon3D: Probing Visual Foundation Models for Complex 3D Scene Understanding

**Yunze Man**[1]  **Shuhong Zheng**[1]  **Zhipeng Bao**[2]

**Martial Hebert**[2]  **Liang-Yan Gui**[1]  **Yu-Xiong Wang**[1]

[1] University of Illinois Urbana-Champaign  [2] Carnegie Mellon University

https://yunzeman.github.io/Lexicon3D

## Abstract

Complex 3D scene understanding has gained increasing attention, with scene encoding strategies built on top of visual foundation models playing a crucial role in this success. However, the optimal scene encoding strategies for various scenarios remain unclear, particularly compared to their image-based counterparts. To address this issue, we present the first comprehensive study that probes various visual encoding models for 3D scene understanding, identifying the strengths and limitations of each model across different scenarios. Our evaluation spans *seven* vision foundation encoders, including image, video, and 3D foundation models. We evaluate these models in *four* tasks: Vision-Language Scene Reasoning, Visual Grounding, Segmentation, and Registration, each focusing on different aspects of scene understanding. Our evaluation yields *key intriguing findings*: Unsupervised image foundation models demonstrate superior overall performance, video models excel in object-level tasks, diffusion models benefit geometric tasks, language-pretrained models show unexpected limitations in language-related tasks, and the mixture-of-vision-expert (MoVE) strategy leads to consistent performance improvement. These insights challenge some conventional understandings, provide novel perspectives on leveraging visual foundation models, and highlight the need for more flexible encoder selection in future vision-language and scene understanding tasks.

## 1 Introduction

Recently, complex 3D scene understanding has emerged as a pivotal area in computer vision, encompassing tasks such as scene generation [25, 26, 27, 34, 77, 96], reasoning [5, 36, 55, 58], and interaction [37, 112]. Leveraging large-scale vision foundation models, many approaches [44, 67, 71, 87, 94] have achieved promising results in various downstream tasks, thereby enabling a wide range of real-world applications, from autonomous driving [57, 78, 82, 117], robotics [60, 112], to multimodal agents [1, 81].

While numerous studies [6, 70, 103] have provided guidance on the use of vision foundation models for 2D image-based tasks, the strategies for 3D scenarios remain unclear. A systematic understanding of complex real-world scenarios involves not only semantic and depth awareness [6], which is possible to evaluate within the 2D domain, but also geometric awareness and the ability to align with multimodal information for reasoning and grounding tasks. To address this gap, our work evaluates the use of different types of visual foundation models for complex scene understanding and seeks to identify the strengths and limitations of each model in different scenarios. Ultimately, this study aims to contribute to the development of more effective and efficient scene understanding systems.

38th Conference on Neural Information Processing Systems (NeurIPS 2024).

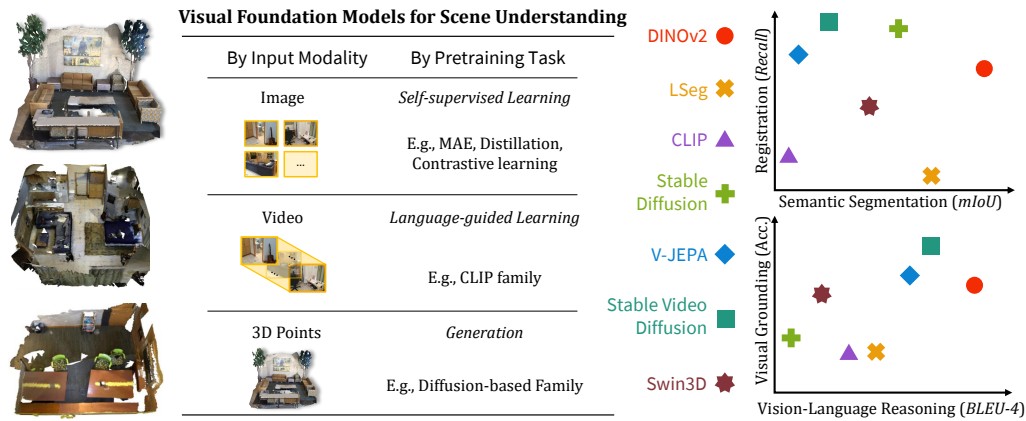

Figure 1: Evaluation settings and major results of different vision foundation models (VFMs) for complex 3D scene understanding. We assess the performance of VFMs on multimodal scene reasoning, grounding, segmentation, and registration tasks.

Concretely, we aim to address several key questions. First, given that most vision foundation models are trained on image or video data, we want to determine *whether 2D foundation models can effectively interpret 3D scenes*. Second, since video models inherently contain temporal information that captures aspects of the 3D structure as well, we investigate *whether they lead to better 3D feature representations compared to image models*. Finally, we seek to identify *the most suitable scenarios for different foundation models trained under various settings*.

To answer these questions, we design a *unified* paradigm to systematically probe visual encoding models for complex 3D scene understanding from different perspectives. Our evaluation spans *seven* vision foundation models in images, videos, and 3D-based models, as shown in Table 1. Our evaluation is conducted among *four* diverse tasks: **Vision-Language Scene Reasoning** assesses the model's ability to reason about scenes based on textual descriptions, evaluating *scene-level* representation; **Visual Grounding** tests the model's capacity to associate language with specific objects within a scene, reflecting *object-level* representation; **Segmentation** evaluates the model's ability to assign semantic labels to each pixel, assessing *semantic* understanding; **Registration** measures the performance of aligning different views of a scene, testing *geometric* capacity. Through these tasks, our aim is to explore the strengths and weaknesses of different vision foundation models in 3D scene understanding, providing insights into their applicability in various scenarios. With the major results demonstrated in Figure 1, our key findings include:

- Image or video foundation models achieve promising results for 3D scene understanding. Among them, DINOv2 [61] demonstrates the best overall performance, showing strong generalizability and flexibility, which is consistent with the observation in 2D scenarios [6]. Our evaluation further verifies its capability in global and object-level 3D vision-language tasks. It can serve as a general backbone for 3D scene understanding.

- Video models, benefiting from temporally continuous input frames, excel in object-level and geometric understanding tasks by distinguishing instances of the same semantics in a scene.

- Visual encoders pretrained with language guidance (*e.g.*, CLIP [68]) *do not* necessarily perform well in other language-related evaluation tasks, challenging the common practice of using such models as default encoders for vision-language reasoning tasks.

- Generative pretrained models, beyond their well-known semantic capacity, also excel in geometrical understanding, offering new possibilities for scene understanding.

- The mixture-of-vision-expert (MoVE) strategies, including combining multi-layer features from the same visual model, and concatenating features from multiple visual models, both lead to a consistent boost of performance across different tasks.

Our work, **Lexicon3D**, provides a unified probing architecture and the first comprehensive evaluation of 3D scene understanding with visual foundation models. The key findings we have achieved above, in conjunction with other interesting observations, suggest exploring more flexible encoder selections in future vision-language tasks to optimize performance and generalization.

| Model | Input Modality | Architecture | Supervision | Dataset |
|---|---|---|---|---|
| DINOv2 [61] | | ViT-L/14 | SSL | LVD-142M |
| LSeg [46] | Image | ViT-L/16 | VLM | LSeg-7Mix |
| CLIP [68] | | ViT-L/14 | VLM | WIT-400M |
| StableDiffusion [73] | | UNet | Generation | LAION |
| V-JEPA [11] | Video | ViT-L/16 | SSL | VideoMix2M |
| StableVideoDiffusion [12] | | UNet | Generation | LVD-F |
| Swin3D [97] | 3D Points | Swin3D-L | Segmentation | Structure3D |

Table 1: Details of the seven evaluated VFMs. In supervision signals, we use "SSL" to represent self-supervised learning, and use "VLM" to represent vision-language modality alignment. A more detailed explanation of the evaluated VFMs is provided in the supplementary material A.

## 2 Related Work

Our work is closely related to methods that focus on extraction of features from images, videos, and 3D assets, as well as learning joint representation spaces for vision-language fusion. A large body of recent literature has explored the representation learning for multimodal visual inputs and their complementary performance in image understanding. In contrast, our study presents a comprehensive analysis of the use of pretrained visual encoders for *zero-shot* 3D scene understanding. *To the best of our knowledge, we are the first to examine pretrained video encoders on 3D scene understanding tasks and to compare image, video, and 3D point encoding strategies in this context.*

**Image self-supervised learning.** In recent years, learning robust and generalizable pretrained image representations has become a prevalent research direction in computer vision and multimodal research. One line of work focuses on learning task-agnostic image features using self-supervised learning (SSL) signals, which include pretext tasks such as colorization [104], inpainting [65], transformation prediction [28], and self-distillation [14, 18, 19, 30, 31]. The recent development of the patch-based image tokenizer, ViT [23], has also led to the emergence of mask autoencoder architectures (MAEs) for feature extraction [8, 32, 115]. Of particular interest, DINOv2 [61], combining a masked-image modeling loss and an invariance-based self-distillation loss, has become one of the most scalable and competitive self-supervised learning architectures that uses only image signals. Another line of work proposes learning image features with text guidance, *i.e.*, using textual descriptions to guide the pretraining of the image encoders [39, 56]. Building upon the powerful image-text encoder CLIP [68], LSeg [46] and BLIP [47, 48] extend the image pretraining objective to more complex visual perception tasks by incorporating pixel-level semantic understanding and encouraging better alignment with large language models (LLMs) [13, 69, 106, 107], respectively.

**Video and 3D representation learning.** Self-supervised representation learning has also been explored in the context of videos and 3D point clouds. Extending the success of the CLIP architecture [68] from images to videos, a body of work proposes to pretrain a video encoder by aligning the feature space with textual guidance extracted from video captions [3, 88, 92, 101]. Other pretext tasks used in video representation learning include next frame prediction [10] and MAE [29, 83, 86]. Among them, V-JEPA [11] adapts the MAE-inspired joint embedding prediction architecture (JEPA) [4, 45] to the spatio-temporal domain, achieving state-of-the-art performance on a wide spectrum of video and image tasks. Despite extensive research on 2D visual foundation encoders, pretrained models for 3D point clouds are significantly fewer due to the lack of large-scale 3D datasets. Existing work has explored contrastive pretraining [38, 91, 109] and masked signal modeling [50, 62, 90, 95, 100, 105] for point representation learning. Recently, benefiting from the rapid advancement of 3D data rendering and large synthetic datasets [21, 113], Swin3D [97] and Uni3D [116] have outperformed other pretraining methods by a significant margin with large-scale pretraining for scene-level perception and object-level understanding, respectively.

**Generation and mixture of experts (MoE) for feature extraction.** With the success of diffusion-based generative models [33, 73, 79], a line of research has begun to explore their role in image perception tasks. These methods extract feature maps or attention maps of a given image from the U-Net architectures of diffusion models and perform various downstream tasks, including depth estimation [24, 74, 111], semantic segmentation [9, 54, 59, 89, 111], object detection [17], and panoptic segmentation [93]. Another line of work [63, 102, 103] investigates the complementary

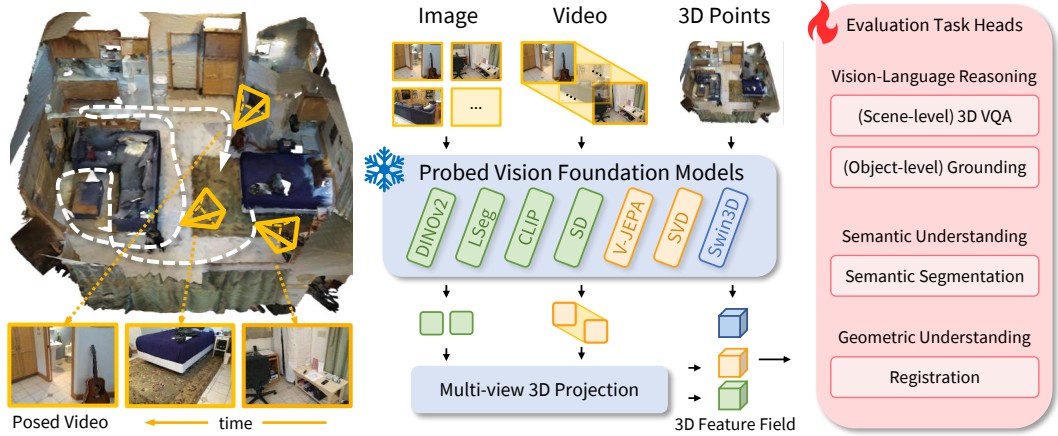

Figure 2: **Our unified probing framework** to evaluate visual foundation models on various tasks.

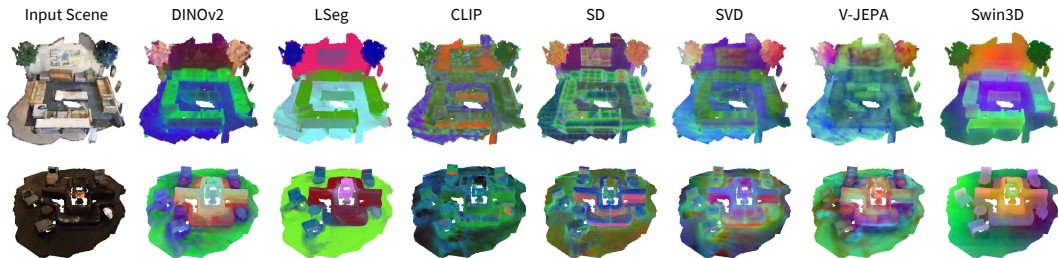

Figure 3: **Visualization of extracted scene features** from different visual foundation models. We use principal component analysis (PCA) to compress the feature embeddings into three dimensions. The clear distinction between colors and patterns demonstrates the behaviors of different models.

nature of different embeddings extracted by multiple foundation backbones and their joint effect on downstream tasks [6, 70]. However, these investigations have been limited to the 2D domain, leaving the potential of leveraging pretrained encoders for perception and reasoning tasks in complex 3D scenes [5, 22, 35, 36, 41, 55, 58, 66, 118] largely unexplored.

## 3    Probing Visual Encoders for Scene Understanding

The objective of Lexicon3D is to evaluate different visual foundation models in complex scene under-standing tasks. We first construct a unified architecture capable of probing different visual foundation models on a spectrum of downstream tasks. Then, we break down the 3D scene understanding task into four sub-tasks, including (1) vision-language reasoning, (2) visual grounding, (3) semantic understanding, and (4) geometric understanding, for a more detailed evaluation.

### 3.1    A Unified Probing Framework

We design a unified framework, as shown in Figure 2, to extract features from different foundation models, construct a 3D feature embedding as scene embeddings, and evaluate them on multiple downstream tasks. For a complex indoor scene, existing work usually represents it with a combination of 2D and 3D modalities. For realistic scenarios [15, 20, 98], videos are usually first captured with handheld cameras and then 3D points are obtained from reconstruction algorithms such as COLMAP [75]. For digital and synthetic scenarios [72, 113], 3D assets are designed and generated first, before images and/or videos are rendered within the created space. Given a complex scene represented in posed images, videos, and 3D point clouds, we extract their feature embeddings with a collection of vision foundation models. For image- and video-based models, we project their features into the 3D space for subsequent 3D scene evaluation tasks with a *multi-view 3D projection module*. Following [22, 35, 36, 66], for a point cloud $\mathbf{P}$, this module produces features $f_{\mathbf{P}}$ for each point $\mathbf{p} \in \mathbf{P}$ given image features $f$ and the pose and camera information $\mathbf{K}, \mathbf{R}$. We first project all points

onto the image plane to obtain their corresponding pixel features. Concretely, for a point $\mathbf{p}$, we obtain its projected pixel $\mathbf{u}$ on the image $i$ with

$$\tilde{\mathbf{u}} = \mathbf{K}_i \mathbf{R}_i \tilde{\mathbf{p}}, \quad \tilde{\mathbf{u}}, \tilde{\mathbf{p}} \text{ represent homogeneous coordinates of } \mathbf{u}, \mathbf{p}, \text{respectively.} \tag{1}$$

In addition, we use an indicator function $\mathcal{I}(\mathbf{p}, i)$ to represent whether a point $\mathbf{p}$ is visible in the image of the $i$-th frame. After finding corresponding pixels of the given point in all image frames, we use mean pooling as an aggregation function $\phi$ to fuse all pixel features to form the point feature $f_{\mathbf{p}}$. Assuming there are M images in total, the projection and aggregation process is represented as:

$$f_{\mathbf{p}} = \phi_{i=1}^{\mathrm{M}}(\mathcal{I}(\mathbf{p}, i) \cdot f_i(\mathbf{K}_i \mathbf{R}_i \tilde{\mathbf{p}})). \tag{2}$$

After projection, we obtain 3D feature fields represented as point cloud feature embeddings for each VFM, and use them as input to the shallow probing heads to evaluate various downstream tasks. To minimize the effect of the model finetuning process, we freeze the parameters for the encoding models to be evaluated, and only tune the linear or shallow probing heads for all tasks.

**Models.** In this work, we focus primarily on evaluating visual foundation models that are frequently leveraged by recent complex scene understanding and multimodal reasoning models. A complex scene can often be represented in posed 2D images and videos or in 3D point clouds. The image and video modalities sacrifice explicit geometry information, but they preserve rich and dense semantic and textural information of a scene. Conversely, the point cloud modality offers the opposite trade-offs. Additionally, the 2D modalities benefit from strong foundation models trained on vast amounts of data, while 3D point backbones only leverage much smaller datasets.

We categorize visual foundation models into three categories, with an overview of the evaluated models provided in Table 1. For image encoders, we evaluate DINOv2 [61], LSeg [46], CLIP [68], and StableDiffusion (SD) [73]. For the video modality, we evaluate V-JEPA [11], the state-of-the-art video understanding model succeeding VideoMAE [83, 86] for a wide spectrum of perception and reasoning tasks, as well as StableVideoDiffusion (SVD) [12], a video generative model. The lack of large-scale 3D scene-level datasets hinders the development of strong zero-shot generalizable 3D foundation models as opposed to their 2D counterparts. However, for comparison, we evaluate Swin3D [97], a 3D backbone that achieves leading performance in zero-shot perception tasks in multiple evaluation datasets compared to previous methods [38, 91, 109]. Swin3D is pretrained on Structured3D [113], a dataset 10 times larger than ScanNet [20]. In addition, we also evaluate the SAM model [43], an open-world instance segmentation model pretrained on the SA-1B [43] dataset, and the Uni3D model [116], which is an object-centric 3D foundation model pretrained on a mixture of datasets proposed by OpenShape [52]. The detailed results of the evaluation of these two models are provided in the supplementary material.

**Feature visualization.** Figure 3 visualizes the features of representative scenes extracted by the vision foundation models. To visualize a high-dimensional feature space with $C$ channels, we apply principal component analysis (PCA) to reduce the feature dimensions to three, normalize them to the range $[0, 1]$, and interpret them as RGB color channels. We demonstrate several representative foundation models' feature visualization, which reveals many intuitive findings. The image models, DINOv2 and LSeg, demonstrate strong semantic understanding, with LSeg exhibiting clearer discrimination due to its pixel-level language semantic guidance. The diffusion-based models, SD and SVD, in addition to their semantic modeling, excel at preserving the local geometry and texture of the scenes because of the generation-guided pretraining. The video models, SVD and V-JEPA, showcase a unique ability to identify different instances of the same semantic concepts, such as the two trees in the first scene and the chairs in both scenes. The 3D model, Swin3D, also exhibits strong semantic understanding. However, due to limited training data and domain shift, its quality is not on par with the image foundation models, despite being pretrained on perfect semantic annotations.

## 3.2 Vision-Language Reasoning

The vision-language reasoning task requires a model to engage in dialogues or answer questions about global understanding and local concepts related to a given complex 3D indoor scene. Following existing methods [36, 112], we formulate this as a visual-question answering (VQA) task using large language models (LLMs) as the backbone – given a 3D scene from multi-view images and point clouds, and a user-prompt question, the LLMs are asked to generate the answer to the question in

| Model | ScanQA (higher means better for all metrics) | | | | | SQA3D (higher means better for all metrics) | | | | |
|---|---|---|---|---|---|---|---|---|---|---|
| | BLEU-1 | BLEU-4 | METEOR | ROUGE | CIDEr | EM-1 | BLEU-1 | METEOR | ROUGE | CIDEr |
| 3D-LLM [36] (for ref.) | 39.3 | 12.0 | 14.5 | 35.7 | 69.4 | 48.1 | 47.3 | 35.2 | 48.6 | 124.5 |
| DINOv2 | 39.2 | 13.4 | 15.3 | 36.8 | 73.2 | 50.1 | 49.5 | 35.6 | 50.7 | 129.1 |
| LSeg | 36.8 | 11.5 | 14.6 | 36.0 | 71.0 | 47.4 | 46.5 | 33.2 | 47.8 | 122.5 |
| CLIP | 36.4 | 10.7 | 14.4 | 36.0 | 70.3 | 48.1 | 47.3 | 34.6 | 48.6 | 124.5 |
| StableDiffusion | 35.5 | 11.7 | 14.1 | 34.9 | 68.2 | 47.7 | 47.2 | 33.6 | 48.3 | 124.0 |
| V-JEPA | 37.4 | 12.1 | 14.7 | 36.7 | 71.4 | 48.4 | 48.1 | 34.8 | 50.0 | 125.7 |
| StableVideoDiffusion | 38.5 | 12.5 | 14.5 | 35.4 | 70.6 | 48.5 | 47.9 | 34.4 | 49.0 | 127.7 |
| Swin3D | 36.1 | 10.5 | 13.9 | 35.4 | 70.0 | 48.3 | 48.0 | 34.1 | 47.3 | 123.9 |

Table 2: Evaluation of vision-language reasoning on ScanQA [5] and SQA3D [55] datasets. The top-2 results for each metric are shown in red and green, respectively. The 3D-LLM results [36] are shown for reference, indicating the relative position of our evaluation results with respect to the leading models trained on this task.

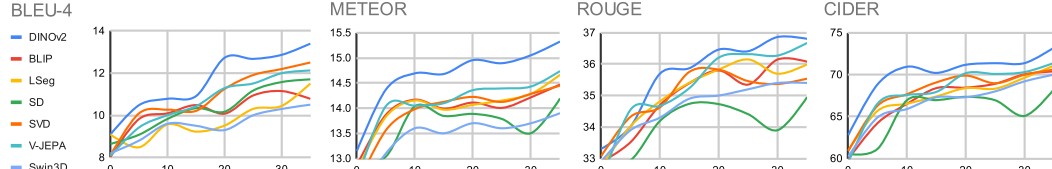

Figure 4: Evaluation curves on the ScanQA benchmark. The $x$-axis demonstrates models trained for different epochs. DINOv2 exhibits clearly superior performance.

an auto-regressive way. This task encompasses universal language-guided reasoning of the complex indoor scene, ranging from global layout to local details.

**Datasets and optimization.** We evaluate the performance on two challenging indoor 3D VQA datasets: ScanQA [5] and SQA3D [55]. Following the evaluation methodology of [5, 36, 55, 58], we report the metrics BLEU [64], ROUGE [49], METEOR [7], and CIDEr [85]. We finetune a Q-Former module [48] to align features from different encoders to the LLM input space. More dataset and optimization details are provided in the supplementary material.

**Evaluation results.** Table 2 and Figure 4 present the results of our evaluation. We observe that image and video encoders generally outperform the 3D point encoder, with DINOv2 achieving the best performance, followed closely by V-JEPA and SVD. Interestingly, we find that for LSeg and CLIP, which are pretrained by language guidance, *their advantage in language alignment does not translate into superior performance on the LLM-guided VQA task.* This finding challenges the common practice of using language-pretrained VFMs [46, 47, 48, 68] as default encoders for LLM-based vision-language reasoning tasks. Instead, it suggests the importance of considering a wider range of encoders, such as DINOv2 and V-JEPA, to support such tasks.

## 3.3 Visual Grounding

Visual grounding is the task of locating an object in a 3D scene based on a text description. Compared to the 3D VQA task, visual grounding places a greater emphasis on object-level reasoning and matching capabilities. The task can be broken down into two sub-tasks: object detection and target discrimination (matching the text description with the target object). Although some methods focus on learning models to tackle both tasks [16, 108], others primarily focus on the discrimination problem [2] by assuming access to ground-truth bounding boxes. For simplicity and to prevent task entanglement, we adopt the latter setting in our evaluation. More specifically, given a 3D scene in the form of multi-view images and point clouds, a free-form language description of objects, and the ground-truth 3D bounding boxes of all objects in the scene, our model's objective is to find the correct objects in the scene that match the language description. We believe that the object detection task requires semantic information from the visual encoder, which is similar in nature to the semantic segmentation task and will be analyzed in Section 3.4.

For the target discrimination task, we first obtain the feature for each object in the scene by taking the average pooling of all points inside its ground truth bounding box. Following Multi3DRefer [108], we use a CLIP text encoder to tokenize the text description, and adopt the attention head in [108] to fuse the text and visual embeddings from the previous steps and output an object score.

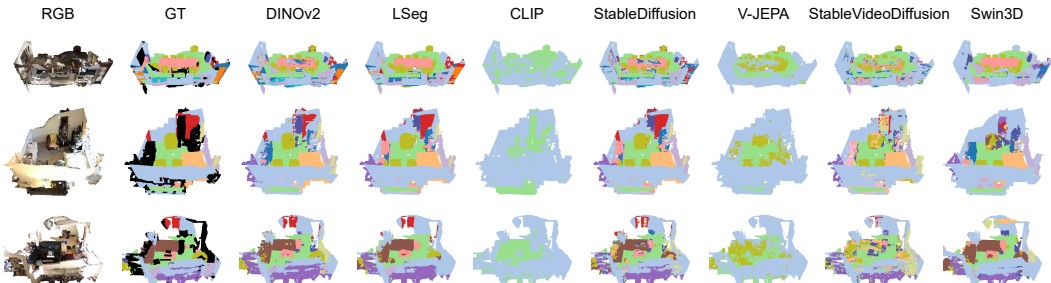

Figure 5: Visualization of 3D semantic segmentation on ScanNet [20]. Image encoders obtain better performance.

**Dataset.** We evaluate on the ScanRefer [16] dataset, which provides 51K text descriptions of 11K objects in 800 ScanNet scenes [20]. We report accuracy for *unique*, *multiple*, and *overall* categories, with *unique* referring to instances that have a unique semantic class in a given scene (easier).

**Optimization.** The model is trained with a cross-entropy loss using the AdamW [53] optimizer following [108]. We train our models for 30 epochs until convergence.

**Evaluation results.** Table 3 presents our results, which show that video encoding models demonstrate significant advantages over image and 3D encoders. The performance gap primarily lies in the *multiple* category, indicating that these models excel at discriminating the correct object among multiple objects of the same semantic category. This capability largely stems from the temporally continuous input frames, which provide instance-aware multi-view consistent guidance. In comparison, the image encoder LSeg, with its language-guided pretraining features aligned with language semantics, can also achieve high accuracy in the *unique* category. However, its performance drops significantly in the *multiple* category.

| Model | Unique ↑ | Multiple ↑ | Overall ↑ |
|---|---|---|---|
| M3DRef [108] (*for ref.*) | 88.0 | 46.1 | 54.3 |
| DINOv2 | 87.0 | 43.4 | 52.0 |
| LSeg | 88.1 | 41.2 | 50.4 |
| CLIP | 86.5 | 41.6 | 50.4 |
| StableDiffusion | 86.4 | 41.9 | 50.6 |
| V-JEPA | 85.6 | 44.9 | 52.9 |
| StableVideoDiffusion | 88.0 | 46.5 | 54.7 |
| Swin3D | 85.7 | 43.2 | 51.6 |

Table 3: Evaluation of 3D object grounding on ScanRefer [16]. Video models exhibit significant advantages.

**Insights from vision-language tasks.** Our evaluation of vision-language reasoning and visual grounding reveals several key findings: (1) The DINOv2 unsupervised image learning model demonstrates strong generalizability and flexibility in global and object-level vision-language tasks. (2) Video encoders benefit from temporally continuous input frames and learn to distinguish instances of the same semantics in a scene, which is highly valuable for object-level understanding tasks. (3) Visual encoders pretrained with language guidance do not necessarily lead to strong performance in other language-related evaluation tasks. These findings suggest exploring a more flexible encoder selection in future vision-language tasks to optimize performance and generalization.

## 3.4 Semantic Segmentation

Semantic segmentation is the task of predicting semantic labels at each 3D position, which requires fine-grained semantic awareness of the scenes. As mentioned in Section 3.1, all types of features are unified in the form of point clouds; therefore, semantic labels are predicted for each point within the point cloud in our setting. More specifically, given a 3D scene in the form of multi-view images and point clouds, the objective in this task is to predict the semantic label for every point in the cloud.

**Dataset.** We conduct the experiments on the ScanNet [20] segmentation dataset which has 1,201 and 312 scenes for training and validation, respectively, with a total of 20 semantic classes for evaluation.

**Optimization.** To make the semantic prediction performance better reflect the fine-grained semantic understanding capability of different features, we use a single linear layer followed by a Sigmoid function to perform a linear probe to predict the probability distribution $\mathbf{y} \in \mathbb{R}^{N \times C}$ for all the labels from the foundation model feature $\mathbf{x} \in \mathbb{R}^{N \times d}$: $\mathbf{y} = \text{Sigmoid}(\text{FC}(\mathbf{x}))$, where $N$ is the number of points in each point cloud, $d$ is the feature dimension, and $C$ is the number of classes for segmentation.

| Model | RR@0.05m (%) ↑ | RR@0.1m (%) ↑ | RR@0.2m (%) ↑ | RRE (°) ↓ | RTE (m) ↓ |
|---|---|---|---|---|---|
| DINOv2 | 82.1 | 93.9 | 96.8 | 1.72 | 0.14 |
| LSeg | 4.8 | 23.7 | 63.8 | 9.80 | 0.59 |
| CLIP | 18.6 | 51.3 | 78.2 | 7.96 | 0.44 |
| StableDiffusion | 91.7 | 96.8 | 98.4 | 1.15 | 0.09 |
| V-JEPA | 90.4 | 96.5 | 99.4 | 1.37 | 0.10 |
| StableVideoDiffusion | 96.8 | 99.0 | 99.7 | 0.83 | 0.06 |
| Swin3D | 60.3 | 81.1 | 91.3 | 3.60 | 0.23 |

Table 5: Evaluation of partial scene registration on ScanNet [20]. We employ Registration Recall (RR) at various RMSE thresholds, Relative Rotation Error (RRE), and Relative Translation Error (RTE) as evaluation metrics. A higher RR indicates better performance, while lower RRE and RTE values signify superior results.

We adopt the standard Adam optimizer [42] with a learning rate of 1e-4 and use a cross-entropy loss to train the linear layer for 20 epochs.

**Evaluation results.** Table 4 and Figure 5 demonstrates that image encoders have better performance than video and 3D encoders on 3D semantic segmentation tasks. The reason is that image encoders like DINOv2 and LSeg gain their semantic awareness during training with contrastive objectives via either SSL or language-driven guidance. In comparison, video encoders have the risk of over-smoothing the multi-view information during multi-frame integration, which may harm the fine-grained semantic understanding capability. As for 3D encoders like Swin3D, the data scarcity in 3D compared to 2D for training the foundation models leads to inferior performance on semantic understanding.

| Model | Acc ↑ | mAcc ↑ | mIoU ↑ |
|---|---|---|---|
| GrowSP [110] (*for ref.*) | 73.5 | 42.6 | 31.6 |
| DINOv2 | 82.5 | 75.4 | 62.8 |
| LSeg | 78.2 | 58.5 | 47.5 |
| CLIP | 39.7 | 7.2 | 3.4 |
| StableDiffusion | 77.2 | 55.5 | 42.6 |
| V-JEPA | 58.7 | 13.2 | 8.1 |
| StableVideoDiffusion | 71.5 | 40.5 | 30.4 |
| Swin3D | 78.0 | 44.8 | 35.2 |

Table 4: Evaluation of semantic segmentation on ScanNet [20] benchmark.

## 3.5 Registration: Geometric Correspondence

To evaluate the geometric information contained in the VFM features, we design the following new task, *partial scene registration*, based on the point cloud registration [51, 99] task that performs homography estimation between two point clouds. From a complete point cloud representing the entire scene, we sample two point clouds $P_1 \in \mathbb{R}^{N_1 \times 3}$ and $P_2 \in \mathbb{R}^{N_2 \times 3}$ within the scene, corresponding to two sets of consecutive viewpoints which have a certain amount of overlapped region but are displaced with a homography transformation. Our goal is to find the homography matrix $H$ that correctly transforms the points in $P_1$ to register with $P_2$. Compared to the semantic segmentation task evaluated in Section 3.4, the partial scene registration task requires the foundation model features to have the capability of finding *geometric correspondence* for registration, which cannot be achieved simply by finding the correspondence according to semantic understanding. For example, in semantic correspondence, we may find two semantically similar points, one on the left side of the sofa in $P_1$, while the other on the right side of the sofa in $P_2$. As a result, if we register the two partial point clouds solely based on semantic correspondence, we will fail to find the correct homography to align one point cloud with the other. The VFMs need to be equipped with geometric understanding capability to achieve decent performance on our partial scene registration task.

**Dataset.** We build our partial scene registration benchmark based on ScanNet [20] dataset. For each scene in ScanNet, we choose views #0 ∼ #31 and views #32 ∼ #63 to render $P_1$ and $P_2$, respectively, so that they can have a certain level of overlap that allows the registration of two partial point clouds. Afterwards, $P_2$ is transformed by a homography $H$ that consists of a rotation $\mathbf{R} \in \mathrm{SO}(3)$ and a translation $\mathbf{t} \in \mathbb{R}^3$. $\mathbf{R}$ is created by a randomly generated quaternion $\mathbf{q} \in \mathbb{R}^4$ for each scene, while each component of $\mathbf{t}$ is randomly sampled from the uniform distribution $[-1.0\mathrm{m}, 1.0\mathrm{m}]$.

**Optimization.** We follow REGTR [99] to adopt a transformer cross-encoder module to enable cross-reasoning of the foundation model features from two point clouds, followed by a lightweight decoder to obtain the corresponding position of every point in the other point cloud for all the $N_1 + N_2$ points in both point clouds, forming altogether $N_1 + N_2$ pairs of correspondences, where $N_1$ and $N_2$ are the number of points in $P_1$ and $P_2$, respectively. Afterward, the rotation $\mathbf{R}$ and the translation $\mathbf{t}$ can be obtained in a closed-form solution solved by a weighted version of the Kabsch-Umeyama [40, 84] algorithm. We use Adam [42] for optimization and train our model for 30 epochs, and follow REGTR

| Model | Time (*sample*) | Time (*scene*) | Mem. |
|---|---|---|---|
| DINOv2 | 25.0 *ms* | 7.5 *sec* | 1.19 G |
| LSeg | 291.2 *ms* | 87.4 *sec* | 2.51 G |
| CLIP | 34.5 *ms* | 10.4 *sec* | 1.19 G |
| StableDiffusion | 42.7 *ms* | 12.8 *sec* | 5.08 G |
| V-JEPA | 175.1 *ms* | 3.3 *sec* | 1.31 G |
| StableVideoDiffusion | 667.1 *ms* | 12.5 *sec* | 11.70 G |
| Swin3D | 937.4 *ms* | 0.9 *sec* | 1.34 G |

Table 6: Complexity analysis of visual foundation models.

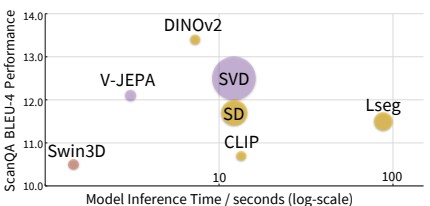

Figure 6: Memory usage of different encoders. An ideal model should be a small circle and be positioned in the upper left.

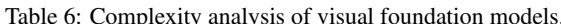

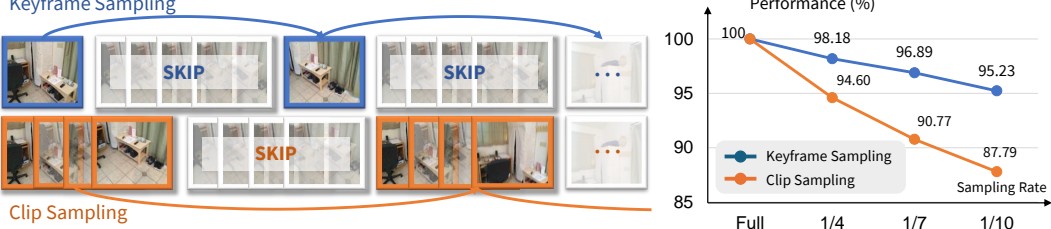

Figure 7: Evaluation on different video downsampling strategies for V-JEPA on the segmentation task. *Keyframe Sampling* samples every $N$ frames to form a new video sequence, while *Clip Sampling* directly samples consecutive video clips. The performance before downsampling is regarded as 100%. Keyframe sampling demonstrates less performance drop with the same level of downsampling.

to adopt Registration Recall (RR), Relative Rotation Error (RRE), and Relative Translation Error (RTE) as evaluation metrics.

**Evaluation results.** Table 5 demonstrates the results for the partial scene registration. We can observe that StableDiffusion and StableVideoDiffusion showcase superior geometric capability in our partial scene registration task. It demonstrates that the pretraining objective of *generation* empowers the foundation models to have a decent capability of finding geometric correspondences in 3D scenes. Another observation is that video encoders generally perform better than image encoders. The reason is that video foundation models have a better understanding of object shapes and geometry within the scenes from the multi-view input frames.

## 4 Analysis

The purpose of this section is to provide additional exploration towards the optimal usage of visual foundation models. The selection of encoding methods requires consideration of the trade-off between memory usage, running time, and performance. We will dive into complexity analysis and the study of design choices for various and a combination of foundation models. More visualization, ablation experiments, and elaboration on the limitations, broader impact, and future direction are presented in the supplementary material.

### 4.1 Complexity Analysis

We compare memory usage, computation time, and model performance (*vision-language reasoning on ScanQA*) in Table 6 and Figure 6. Our findings show that image encoders generally require less time to process a sample compared to video and 3D encoders. And diffusion-based models, when used for feature extraction, require significantly more memory than other discriminative models. However, the drawbacks in running time become evident for 2D backbones, especially image encoders, when attempting to obtain a scene embedding by aggregating multi-view image embeddings. To illustrate this, we consider a 300-frame video as an exemplar of posed 2D information for a complex scene (a 10-second video at 30 FPS). As the length of the video increases, 2D methods, which necessitate feature extraction for each image frame, rapidly consume a substantial amount of time to process a single scene. In contrast, a 3D point encoder requires significantly less time to process a scene. Nevertheless, 3D encoders exhibit relatively poor model performance, which can be attributed to the scarcity of training data. To fully demonstrate their potential in scene understanding tasks, efforts should be directed toward enhancing the generalizability of 3D foundation models. All analyses and computations are performed on an NVIDIA A100 GPU.

| Stable Diffusion | BLEU-1↑ | BLEU-4↑ | METEOR↑ | ROUGE↑ | CIDEr↑ |
|---|---|---|---|---|---|
| *Evaluation of noise level t* | | | | | |
| $t = 1$ *step* | 35.3 | 11.6 | 14.0 | 34.5 | 68.5 |
| $t = 25$ *steps* | 35.6 | 11.5 | 14.0 | 34.2 | 68.3 |
| **t = 100 *steps*** | 35.5 | 11.7 | 14.1 | 34.9 | 68.2 |
| $t = 200$ *steps* | 34.3 | 10.9 | 13.9 | 33.9 | 66.6 |
| *Evaluation of feature layer l* | | | | | |
| $l = 0$ | 33.6 | 10.5 | 13.3 | 32.6 | 65.9 |
| **l = 1** | 35.5 | 11.7 | 14.1 | 34.9 | 68.2 |
| $l = 2$ | 34.9 | 11.4 | 14.0 | 34.5 | 68.0 |

Table 7: Evaluation of diffusion noise level and feature layers when using StableDiffusion [73] for feature extraction. The settings we choose are highlighted with the grey color.

## 4.2 Ablation Study – Insights into Optimal Usage of Visual Foundation Models

**Video downsampling strategy.** Long and high frame-per-second videos take a lot of space to store and time to process. We explore two straightforward ways of conducting temporal downsampling to achieve more efficient processing without sacrificing too much performance. As shown in Figure 7, we explore the *keyframe sampling* (blue) and *clip sampling* (orange) strategies. We can observe that keyframe sampling is a better strategy than clip sampling in this setting, more wisely balancing the trade-off between video processing overhead and task performance.

**Combination of multiple encoders.** We explore whether a mixture of foundation models (experts) has the potential to strengthen the capability of 3D scene understanding. We experiment on the 3D semantic segmentation task with three feature sources: LSeg, StableDiffusion, and Swin3D. When combining different feature sources, we concatenate all features along the channel dimension for every point in the point cloud. The results are shown in Figure 8. After combining features from different sources, there exists a potential that the semantic understanding capability can be boosted in a *mixture of experts* manner. However, it is not necessarily true that combining the best features will lead to the best performance. For example, LSeg **(1)** has stronger capability on semantic segmentation than StableDiffusion **(2)** and Swin3D **(3)** individually, but it is StableDiffusion + Swin3D **(2+3)** that reaches the best performance when combining two features together.

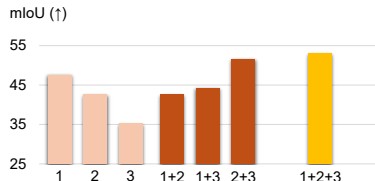

Figure 8: Evaluation on the segmentation task with **(1)** LSeg, **(2)** SD, **(3)** Swin3D, and their combinations.

## 4.3 Diffusion Noise Level and Feature Layer

In Table 7, we evaluate the effect of different noise level (*noise steps*) and different feature layers in the decoder module in leveraging StableDiffusion (SD) [73] for feature extraction. The results show that for SD, adding noise $t < 100$ steps in general leads to the best performance. When $t$ increases beyond 100 steps, the performance starts to downgrade. As for decoder layers, the decoding portion of the UNet consists of 4 blocks. We skip the final layer closest to the output and consider layers 0, 1, and 2. The results demonstrate that the output features of the layer one decoder lead to the best performance. These observations are consistent with the study in [6, 103].

## 5 Conclusion

This paper presents the first comprehensive analysis of leveraging visual foundation models for complex 3D scene understanding. We explore the strengths and weaknesses of models designed for various modalities and trained with different objectives. Our study reveals the superior performance of DINOv2, the advantages of video models in object-level tasks, and the benefits of diffusion models in geometric registration tasks. Surprisingly, we find limitations of language-pretrained models in language-related tasks. The extensive analysis suggests that a more flexible encoder selection and fusion can play a crucial role in future scene understanding and multimodal reasoning tasks.

## Acknowledgments

This work was supported in part by NSF Grant 2106825, NIFA Award 2020-67021-32799, the IBM-Illinois Discovery Accelerator Institute, the Toyota Research Institute, and the Jump ARCHES endowment through the Health Care Engineering Systems Center at Illinois and the OSF Foundation. This work used computational resources on NCSA Delta through allocations CIS220014, CIS230012, and CIS230013 from the Advanced Cyberinfrastructure Coordination Ecosystem: Services & Support (ACCESS) program, and on TACC Frontera through the National Artificial Intelligence Research Resource (NAIRR) Pilot.

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

# Lexicon3D: Probing Visual Foundation Models for Complex 3D Scene Understanding

## Supplementary Material

## A    Additional Experiment Details

In this section, we provide a detailed introduction of all the visual foundation models we have evaluated, including the checkpoints we use and how we extract feature representations from the encoder backbones.

### A.1    Evaluated Visual Foundation Models

Our evaluation and analysis are conducted mainly on the seven models listed in Table 1 in the main paper. We have chosen models such that they cover most of the backbones used by recent 3D scene understanding and reasoning work. In this part, we discuss all the models we have used in our experiments and explain their pretraining objective, the dataset used for pretraining, the public checkpoints we choose, and the method we leverage to extract features from their backbones. We start with image foundation models, and then video and 3D models.

**DINOv2 [61].** DINOv2 leverages an image-wise contrastive objective by minimizing the distance of features from the same samples and maximizing those from different samples. It also includes a patch-wise denoising objective by performing a reconstruction from masked inputs. It is trained on a large-scale image dataset, LVD-142M [61], which contains 142 million unlabeled images. We take the standard DINOv2 implementation[*] and use the pretrained ViT-L/14 checkpoint for our evaluations.

**LSeg [46].** LSeg aims to align visual features from images with the corresponding semantic information provided by natural language descriptions by maximizing the correlation between the text embedding and the image pixel embedding of the ground truth class of the pixel. We use the official checkpoint[†] of ViT-L/16 that is trained on a mixture of seven datasets [46].

**CLIP[68].** CLIP aligns visual and textual representations in a shared embedding space through contrastive learning by maximizing the similarity between the embeddings of corresponding image-caption pairs while minimizing the similarity of non-matching pairs. CLIP is trained on a large and diverse dataset of image-caption pairs sourced from the Internet including over 400 million image-text pairs. We use the official implementation and checkpoint[‡] with ViT-L/14 as the backbone for our evaluation.

**StableDiffusion (SD) [73].** SD is a diffusion-based model used for generating high-quality images from text prompts. The model is trained to gradually remove noise from images, transforming random noise into coherent images that match the provided text descriptions. It is trained on LAION5B [76] which contains over five billion of images paired with detailed captions. We follow DIFT [80][§] to extract features from SD and we use the SD2.1 checkpoint for our evaluation. We use the features from block index 1 for all tasks. The noise timestep is set to 100 by default. We use null-prompt as the text condition.

**StableVideoDiffusion (SVD) [12].** SVD is an extension of SD from image generation to video generation by incorporating additional temporal modules. SVD is first initialized from an image-level pretrained diffusion checkpoint (SD2.1), and then further finetuned on 10 million videos. We use their publicly released image-to-video variant (SVD-xt) [¶]. We build our feature extractor pipeline following DIFT [80] and extract the features from block index 1 for all tasks. The noise timestep is set to 25 by default. We use the first-frame image as the condition for all the cross-attention modules, while we use the unconditional version for the latent input of the UNet – we concatenate an all-zero

---

[*] https://github.com/facebookresearch/dinov2
[†] https://github.com/isl-org/lang-seg
[‡] https://github.com/openai/CLIP
[§] https://github.com/Tsingularity/dift
[¶] https://huggingface.co/stabilityai/stable-video-diffusion-img2vid-xt

vector to the framewise embeddings. Each time, we feed a 25-frame video clip to SVD to process the features.

**V-JEPA [11].** V-JEPA aims to learn robust visual representations by predicting future states of visual data. This model is pretrained on a mixed video dataset containing more than 2 million videos. We take their official implementation[‖] and the ViT-L/16 checkpoint with a resolution of $224 \times 224$. We obtain their per-patch representation by removing the last pooling and linear layers. Each time, we feed a 16-frame video clip to V-JEPA to process the features.

**Swin3D [97].** Swin3D adapts the Swin Transformer to handle 3D data, such as point clouds and volumetric data. We use the official checkpoint[**] that takes Swin3D-L as the backbone and is pretrained using the Structure3D dataset [114] with semantic segmentation as the target.

### A.2 Additional Evaluation Details for Vision-Language Scene Reasoning

**Datasets.** We evaluate the performance on two challenging indoor 3D VQA datasets: ScanQA [5] and SQA3D [55]. SQA3D features over 33K QA pairs, while ScanQA consists of more than 41K pairs. Each entry in these datasets includes a complex 3D indoor scene, a question, and the corresponding answers. We use the splits provided by the respective datasets.

**Optimization.** We keep the LLM parameters frozen and finetune the shallow visual projection Q-Former module [48] to align the features of different encoders with the LLM input space. Unlike 3D-LLM [36], we train the Q-Former module from scratch for a fair comparison of all encoders. Following the approach of 3D-LLM, we pretrain the module for 10 epochs using the 3D-Language dataset [36] and then finetune it on the training split of the two evaluation datasets for 35 epochs. Both stages use the AdamW [53] optimizer with a linear warm-up and cosine decay learning rate scheduler. Although longer training can further improve performance, trends stabilize after 35 training epochs.

### A.3 Additional Evaluation Details for Registration

**Dataset generation.** When generating the corresponding partial scene point clouds from the ScanNet dataset, due to memory constraint, we downsample the partial scene point clouds to 4,096 points each with the farthest point sampling (FPS) algorithm, if the number of points in $P_1$ and $P_2$ is greater than 4,096. We follow the same train/val split on the semantic segmentation task in our partial scene registration task.

### A.4 License of Datasets Used

We list the licenses of all the datasets we have used during our evaluation:

- ScanNet [20]: MIT License.
- ScanQA [5]: CC BY-NC-SA 3.0 License.
- SQA3D [55]: CC-BY-4.0 License.
- ScanRefer [16]: CC BY-NC-SA 3.0 License.
- 3D-Language-Data [36]: MIT License.

In addition, we utilize a number of public foundation model checkpoints pretrained on various data sources in our paper. Please refer to their original paper for the license of datasets they have used in pretraining their models.

## B Additional Experimental Results

### B.1 Comparison Between Scene-level and Object-centric Models

Uni3D [116] is a general transformer-based 3D foundation model pretrained on a mixture of four object-centric datasets [52]. Focusing on object-centric understanding, it has a restriction on the

---

[‖]https://github.com/facebookresearch/jepa
[**]https://github.com/microsoft/Swin3D

| Model | VQA (CIDEr) ↑ | Grounding (Acc.) ↑ | Segmentation (mIoU) ↑ | Registration (RTE) ↓ |
|---|---|---|---|---|
| Swin3D | **70.9** | **51.6** | **35.2** | 0.23 |
| Uni3D | 63.1 | 51.1 | 2.7 | **0.08** |

Table A: Comparison between Uni3D and Swin3D on four of our evaluation tasks. Object-centric and scene-centric methods demonstrate significant differences.

| Model | VQA (CIDEr) ↑ | Grounding (Acc.) ↑ | Segmentation (mIoU) ↑ | Registration (RTE) ↓ |
|---|---|---|---|---|
| LSeg | **71.0** | **50.4** | **47.5** | 0.59 |
| SAM | 68.6 | 50.1 | 30.9 | **0.09** |

Table B: Comparison between SAM and LSeg on four of our evaluation tasks. Instance-aware segmentation and semantic-aware segmentation methods demonstrate significant differences.

number of input points and output dimensions. In contrast, Swin3D [97] is pretrained on a smaller scene-level dataset [113], but is designed to focus more on understanding scene-level information. To demonstrate the performance of Uni3D[††], we conduct experiments with its features on our evaluation benchmarks. More specifically, following the part segmentation details in Uni3D's appendix (Sec. B), we use Uni3D-giant, selecting features from the 16th, 28th, and 40th (last) layers to form grouped point patches. We then employ PointNet++'s [91] feature propagation to up-sample group features into point-wise features. It is worth noting that Uni3D's ScanNet visualizations in their paper were achieved by applying Uni3D to each object instance based on ground truth instance segmentation, not by direct application to the whole scene.

The results are shown in Table A. From the table we have several interesting findings:

- For **scene-level tasks** (3D VQA and Semantic Segmentation): Uni3D underperforms the scene-level pretrained Swin3D model. This is likely due to the object-centric pretraining recipe of Uni3D, causing the failure of feature extraction on large scenes with orders of magnitude more points than single objects.

- For **object-centric tasks** (3D object grounding): Uni3D achieves comparable results with Swin3D. However, some grounding questions require not only object-level semantics, but also inter-object relationship and global room information, which Uni3D lacks. We believe that combining object-centric and scene-level representations would be an impactful future direction to achieve better object grounding in complex 3D scenes.

- For **geometric understanding task** (Registration): Uni3D achieves better performance than Swin3D, suggesting that geometric knowledge from object-centric pretraining generalizes well to scene-level geometric matching, especially given the task's use of downsampled partial scenes bridging the distribution gap between object-level and scene-level point clouds.

### B.2 Comparison Between Semantic and Instance Segmentation Models

Segment Anthing Model (SAM) [43] is an open-world instance segmentation model pretrained on a very large dataset SA-1B [43]. In Table B, we compare the performance of SAM with LSeg [46], a semantic segmentation model. For SAM, we use the official pretrained model checkpoint with ViT-L as the backbone encoder, matching the model size with other visual foundation models in our experiments.

With the results in Table B, we offer the following analysis:

- First, it is crucial to highlight the fundamental differences between LSeg and SAM. LSeg is designed to perform language-driven *semantic* image segmentation, providing semantic-aware representations. In contrast, SAM is primarily an *instance* segmentation model that focuses on local representations and excels in detecting edges. These distinctions result in varied performance on the four tasks in our evaluation.

- Among the four tasks, *3D VQA* and *semantic segmentation* require a deep semantic understanding of the 3D scenes, where LSeg naturally outperforms SAM. For *3D grounding*, both

---

[††]https://github.com/baaivision/Uni3D

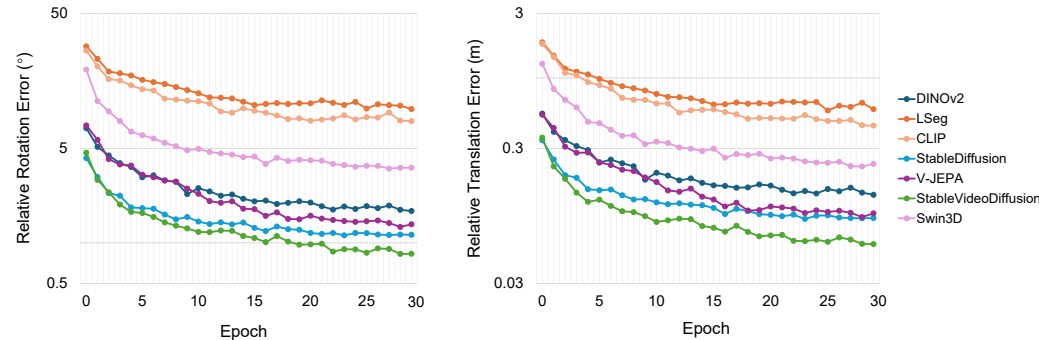

Figure A: Evaluation curves of Relative Rotation Error (RRE) and Relative Translation Error (RTE) on the partial scene registration task during different training stages.

semantic and spatial understanding is necessary; therefore, LSeg and SAM exhibit similar performance in this task. The *registration* task, however, demands matching point clouds using distinguishable local features. Here, SAM's ability to provide precise local features positions it as a strong performer in this geometry-oriented task

- Overall, SAM is not well-suited for numerous downstream tasks, particularly those requiring semantic comprehension. This conclusion is consistent with the previous study [70]. However, we also reveal that it excels in tasks that benefit from robust local feature representation.

### B.3 Evaluation Curves During Different Training Stages

We show the evaluation curves for the partial scene registration task in Figure A. We can observe that the performance ranking of different foundation models stays mainly unchanged throughout the training process.

### B.4 Additional Qualitative Results

We show additional qualitative results for partial scene registration in Figure B, demonstrating that the family of StableDiffusion and StableVideoDiffusion which use the objective of generative pretraining obtains superior performance. In addition, video encoders like V-JEPA and StableVideoDiffusion are equipped with a stronger capability to find geometric correspondences.

## C   Limitations and Future Work

Although we have made a substantial effort to explore the role of visual foundation models in various scene understanding tasks, our perception of this problem remains relatively limited. This section provides a detailed discussion of the limitations and outlines potential future directions.

**Model capacities are not strictly identical or comparable.**   Our evaluation focuses on seven vision foundation models due to their availability and common use in recent work. Consequently, all our experiments are based on publicly available checkpoints. Although we have attempted to choose models with similar capacities, achieving strictly identical backbone architectures was not possible without re-training all the baselines ourselves. However, such experiments require an enormous amount of computational resources that we cannot afford.

**Our evaluation focuses on indoor scenarios.**   Recent literature often separates the study of perception and reasoning of indoor scenes from outdoor scenarios, which are often relevant to autonomous driving or robotics applications. Outdoor scenarios present different challenges compared to indoor scenes. Lexicon3D focuses its evaluation solely on indoor scenes. While this is a valid choice considering that most scene-level multimodal benchmarks are still based on indoor scenes, it is not comprehensive. Outdoor scenarios contain large ego-movement speeds and many more dynamic moving objects than indoor scenes. Evaluating these scenes will likely lead to unique observations, and we consider this a direct future direction.

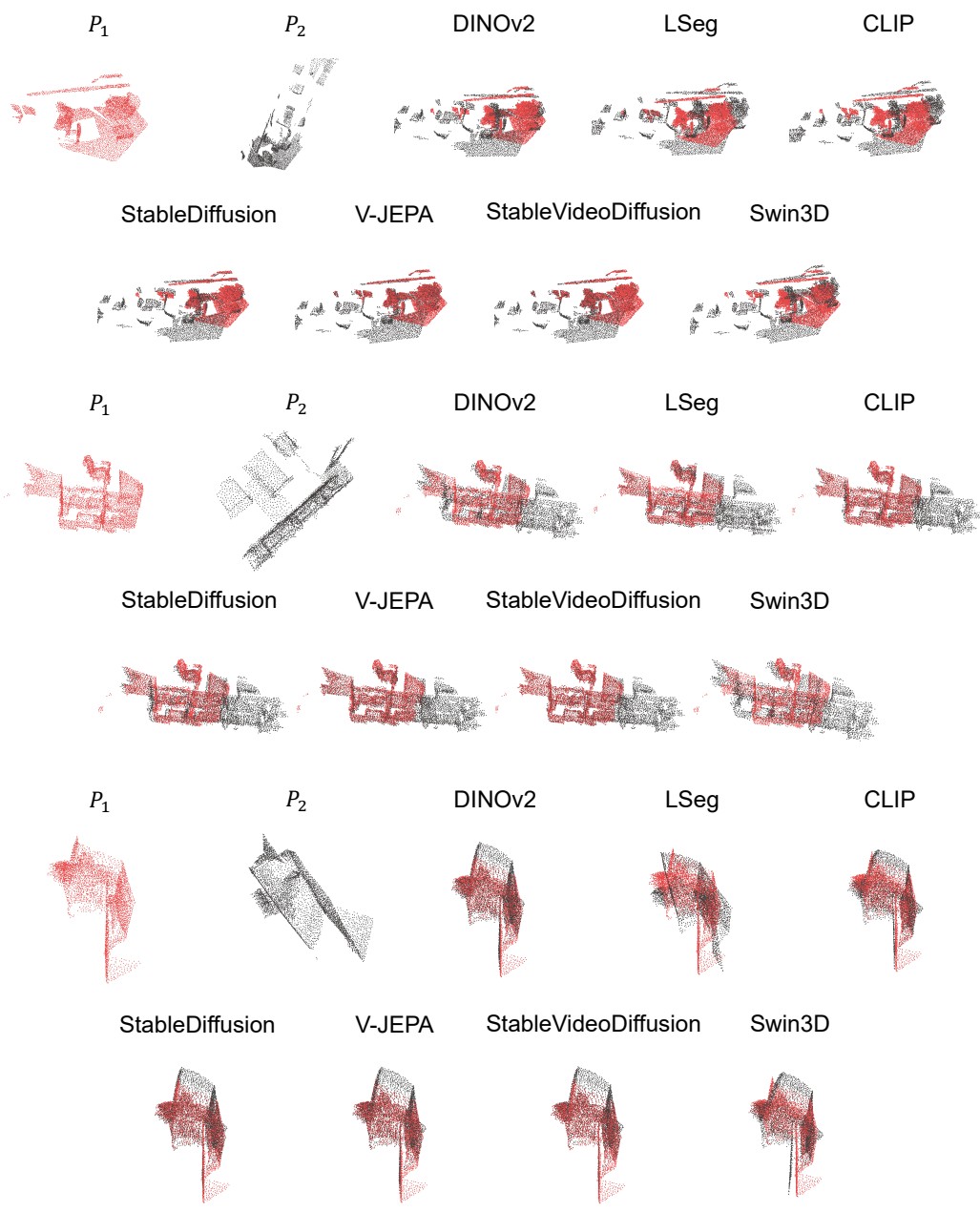

Figure B: Visualization of partial scene registration results. The StableDiffusion and StableVideoDiffusion family of generative models receives superior performance. In addition, video encoders such as V-JEPA and StableVideoDiffusion have better geometric understanding capability than image encoders.

**We adopt the most straightforward approach to probing.** To evaluate the capabilities of the visual foundation models, we freeze their parameters and only tune the linear or shallow probing head. This approach allows us to analyze the capabilities of the pretrained methods without altering their models through the finetuning process. Although we argue that probing the frozen encoder provides the most accurate understanding of these models, we acknowledge that the ability to quickly adapt to new tasks with finetuning is also an important aspect of an encoder. However, finetuning these large-scale models, which often have close to billion-level parameters, requires a significant amount of time and computational resources. We leave this study for future work.

# D Societal Impact

We anticipate a potential positive societal impact from our work. Lexicon3D represents one of the first steps towards a comprehensive understanding of large-scale visual foundation models in real-world 3D scene analysis and reasoning. This understanding could lead to the development of more robust and efficient scene encoding systems, which benefit a wide range of applications, including autonomous driving, virtual reality, household robots, and multimodal chatbots. Ultimately, this could contribute to a more inclusive, efficient, and safer world, where technology understands and adapts to the diverse ways humans perceive and navigate their environments.

**Potential negative societal impact.** We do not see a direct negative societal impact on our work. Indirect potential negative impact involves misusing strong scene-encoding foundation models for surveillance or virtual reality. We believe that it is crucial for researchers to proactively consider these concerns and establish guidelines to ensure responsible usage of these models.

