# OpenReview forum: "Lexicon3D: Probing Visual Foundation Models for Complex 3D Scene Understanding"
_NeurIPS.cc/2024/Conference — NeurIPS 2024 poster_

### Official Review · Reviewer_vFsT · 2024-07-01

**Soundness:** 3
**Presentation:** 4
**Contribution:** 3
**Rating:** 6
**Confidence:** 4

**Summary:**

This paper presents a comprehensive analysis of leveraging visual foundation models for complex 3D scene understanding. The authors unify three kinds of feature representation: image, video and 3D in a unified paradigm and analyze the effectiveness of those representations on different kinds of 3D tasks.

**Strengths:**

1. The study is thorough and meaningful. I think it provides interesting finds to the 3D vision community.
2. Experiments are extensive and comprehensive. The chosen tasks and methods are representative.

**Weaknesses:**

1. This work lacks a unified conclusion. The experimental observations are independent (in some extent like an experimental report rather than a research paper). It is better for the authors to summarize the core underlying principles, or design some models according to the observation, which may make this work more applicable.
2. Since indoor 3D perception is mainly applied in embodied AI system, I suggest the author further study 3D scene understanding in an online setting [1, 2] rather than the current offline setting, which can be directly adopted in real-world robotic tasks.

[1] Fusion-aware point convolution for online semantic 3d scene segmentation, CVPR 2020
[2] Memory-based Adapters for Online 3D Scene Perception, CVPR 2024

**Questions:**

No

**Limitations:**

Limitation is well addressed.

---

> ### Author Rebuttal · Authors · 2024-08-07
>
> We appreciate your comments, and address your concerns as follows:
> ***
> 1. *Q: This work lacks a unified conclusion. The experimental observations are independent (to some extent like an experimental report rather than a research paper). It is better for the authors to summarize the core underlying principles, or design some models according to the observation, which may make this work more applicable.*
>
> We appreciate the reviewer's suggestion. We would like to highlight that our work presents a **unified and principled evaluation and probing framework**, emphasizing simplicity and generalizability. Hence, the conclusions we arrive at are accurate and universal. Empirically, we did not observe a single unified conclusion: for example, no single VFM can uniformly dominate all visual tasks. However, we think this diversity in performance aligns with the varied pretraining tasks and input modalities of these 2D foundation models.
>
> To address the reviewer's concern, we summarize our **key findings as general principles**:
> * Leveraging 2D VFMs in 3D scene perception and multi-modal reasoning tasks consistently yields significant improvements.
> * Pretraining tasks and input modality have significant influence over a foundation model’s strengths and weaknesses, which can be effectively interpreted with our simple and unified probing framework.
> * The straightforward yet efficient concatenation-based Mixture-of-Vision-Experts (MoVE) effectively leverages complementary knowledge from different VFMs.
>
> These principles demonstrate the applicability of our work and provide valuable insights for future research and applications in the field. They guide the selection of appropriate VFMs for specific tasks and highlight areas for improvement in existing models. Our comprehensive evaluation framework, while yielding diverse results, offers a unified approach to understanding and comparing VFMs in 3D vision tasks.
>
> ***
> 2. *Q: Since indoor 3D perception is mainly applied in embodied AI systems, I suggest the author further study 3D scene understanding in an online setting [A, B] rather than the current offline setting, which can be directly adopted in real-world robotic tasks.*
>
> We appreciate the reviewer's suggestion to explore online 3D scene understanding. We will include the discussion of these works in the revised manuscript.
>
> * First, we would like to emphasize that our current exploration in the offline setting holds significant value.
>     * It provides a clean, simplified scenario suitable for pure evaluation of visual embedding qualities.
>     * To our knowledge, there hasn't been a systematic exploration under the offline setting in the community until now.
> * Regarding the interesting online setting, based on insights from our offline evaluation, we can offer a few insights and some preliminary thought experiments:
>     * 2D visual foundation models (VFMs), especially video-based ones, are likely to excel in online scenarios, as most perception devices capture video modality rather than point clouds directly.
>     * Inference time would become a crucial metric for selecting VFMs. For high-resolution or high-frame-rate videos, acceleration methods like key-frame selection may be necessary to ensure timely computation.
>     * For extremely long videos, memory-bank or feature compression techniques might be required to optimize storage.
> * While the online setting is an intriguing direction, it lies beyond the scope of our current submission. Due to time constraints during the rebuttal phase, we consider this a promising avenue for future work, building upon the insights gained from our offline analysis.

---

> > ### Comment · Area_Chair_2Gix · 2024-08-08
> >
> > Dear reviewer,
> > The author-reviewer interaction period has started. Please read the responses provided by the authors, respond to them early on in the discussion, and discuss points of disagreement.
> > Thanks

---

> > ### Comment · Reviewer_vFsT · 2024-08-08
> >
> > The authors' rebuttal has solved most of my concerns. I will raise the score from 5 to 6.

---

> > > ### Author Response · Authors · 2024-08-08
> > > **Thank you for your positive feedback!**
> > >
> > > We appreciate the reviewer for the positive feedback. Your constructive comments and suggestions are indeed helpful for improving the paper. Also, many thanks for raising the score.
> > >
> > > We will continue to improve our work and release the code. If the reviewer has any follow-up questions, we are happy to discuss!

---

### Official Review · Reviewer_yLXP · 2024-07-10

**Soundness:** 2
**Presentation:** 3
**Contribution:** 1
**Rating:** 5
**Confidence:** 4

**Summary:**

This paper explores the importance of scene encoding strategies in the context of 3D scene understanding, an area gaining significant attention. The authors investigate the optimal encoding methods for various scenarios, addressing the lack of clarity compared to image-based approaches. They conduct an in-depth study of different visual encoding models, evaluating their strengths and weaknesses across multiple scenarios. The study examines seven foundational vision encoders, including image-based, video-based, and 3D models, across four key tasks: Vision-Language Scene Reasoning, Visual Grounding, Segmentation, and Registration. The findings reveal that DINOv2 performs exceptionally well overall, video models are particularly effective for object-level tasks, diffusion models excel in geometric tasks, and language-pretrained models have unexpected limitations in language-related tasks. These results challenge existing assumptions, provide fresh insights into the use of visual foundation models, and underscore the need for adaptable encoder selection in future vision-language and scene-understanding research.

**Strengths:**

1, The paper is well-written and easy to understand.
2, It surveys most of the current visual foundation models.

**Weaknesses:**

1, I do not see any new insights from this work. Numerous previous studies [1, 2, 3] have already demonstrated that leveraging foundation models can improve 3D understanding.

2, This work simply projects visual foundation models from different views onto the 3D point cloud and fine-tunes them. There is no specific design to better integrate/distill these features.

3, The latest 3D foundation model, Uni3D, is not discussed in this work.

4, Inference time is a significant issue, as the input requires multiple-view images.

[1] Multi-View Representation is What You Need for Point-Cloud Pre-Training
[2] Bridging the Domain Gap: Self-Supervised 3D Scene Understanding with Foundation Models
[3] CLIP2Scene: Towards Label-efficient 3D Scene Understanding by CLIP
[4] Uni3D: Exploring Unified 3D Representation at Scale

**Questions:**

See weakness.

**Limitations:**

Yes

---

> ### Author Rebuttal · Authors · 2024-08-07
>
> We appreciate your comments, and address your concerns as follows:
> ***
> 1. *Q: New insights from this work? Numerous previous studies [A, B, C] have demonstrated that leveraging foundation models can improve 3D understanding.*
>
> We appreciate the references provided, and will include the discussion of these methods in the revised manuscript. We would like to emphasize that **our work aims to comprehensively understand the strengths and limitations of a large group of VFMs in various 3D scenarios**, rather than to merely “*demonstrate that leveraging VFMs can improve 3D perception*”. The unique novelty of our work lies in:
> * **Scope**: Unlike these prior methods [A,B,C] that focus on improving specific perception tasks (detection and segmentation) using a limited set of models (usually CLIP and DINOv2), our work is the first systematic study of a broad range of VFMs across diverse tasks, including 3D perception, shape registration, and multi-modal grounding and reasoning. These VFMs (CLIP, DINOv2, StableDiffusion, LSeg, StableVideoDiffusion, V-JEPA, Swin3D) are pretrained on diverse data using different objectives, and some of them, such as the image/video diffusion-based ones are never explored as visual encoders for 3D understanding.
> * **Objective**: Our primary aim is to comprehensively understand the strengths and limitations of different VFMs, rather than merely improving performance on downstream tasks. As mentioned by Reviewers **vFsT** and **4P6H**, “the insights and findings from the analysis are meaningful and crucial for the 3D vision and language community.”
> * **Novel findings**: Our analysis reveals several new insights, as discussed in Lines 50-61, including
>     * The effectiveness of video foundation models in object-level tasks (Section 3.3)
>     * Previously overlooked limitations of language-pretrained models (Sections 3.2, 3.3)
>     * Advantages of generative-pretrained models in geometric tasks (Section 3.4)
>
> These findings demonstrate the uniqueness of our work and provide valuable insights for future research and applications in the field. They guide the selection of appropriate VFMs for specific tasks and areas for improvement in existing models, which are not provided in the study of [A,B,C].
>
> ***
> 2. *Q: This work simply projects visual foundation models from different views onto the 3D point cloud and fine-tunes them. There is no specific design to better integrate/distill these features.*
>
> Our approach is intentionally designed to be simple and straightforward. It is the simplicity of our probing framework that makes our evaluation generalizable. More specifically, we choose this simple design for these reasons:
> * **Focus on VFM comparison**: Our primary goal is to analyze and compare different VFMs, rather than to propose new architectures or improve task performance. A simple architecture allows us to minimize confounding factors and focus on the intrinsic capabilities of the VFMs. In comparison, having a more advanced integration or distillation design would introduce more entanglement in the evaluation. But we agree with the reviewer that a more advanced design can be direct future work based on our insights.
> * **Consistent with prior work**: This approach aligns with recent studies like Probe3D (CVPR 2024, [6]) and ATO2F (NeurIPS 2023, [97]), which also employ simple architectures to prioritize model analysis. Compared with these methods, our work performs investigations on a broader range of foundation models, and is the first to systematically study them in 3D scene-level multi-modal tasks.
> * **Effectiveness of simple integration**: Despite its simplicity, our Mixture-of-Experts approach using straightforward concatenation demonstrates significant performance improvements, as evidenced in Section 4.2 and Figure 8.
>
> ***
> 3. *Q: The latest 3D foundation model Uni3D is not discussed.*
>
> We will include the discussion of this method in the revised manuscript. While Uni3D is a general transformer-based 3D VFM, it is pretrained on object-centric datasets (Objaverse, ShapeNet, etc.), with a restriction of input and output dimensions. In contrast, our research focuses on more challenging and universal scene-level understanding.
>
> **To demonstrate the performance of Uni3D**, we conduct experiments with its features on our evaluation benchmarks. Due to the restriction of rebuttal length, we include the results and observations in the **General Response**. Please refer to that part for more details.
>
> ***
> 4. *Q: Inference time is a significant issue, as the input requires multiple-view images.*
>
> * **Study Objective**: Our study's primary aim is to provide a simple, unified framework for evaluating and analyzing different VFMs, rather than proposing a new method optimized for low latency. Our analysis, particularly in Table 6 and Figure 6, clearly illustrates the trade-off between performance and inference time in existing methods.
> * **Model-Dependent Inference Time**: The inference time of our evaluation method depends on the specific type of VFMs we use, rather than our probing framework. For example, when proving the Swin3D model, we directly take 3D point clouds as input, rather than multi-view images, leading to faster inference time.
> * Regarding inference time of multi-view images, our work also reveals several key insights:
>     * For scenes captured by a large number of multi-view images (long videos), video foundation models like V-JEPA can be more efficient than single-frame models.
>     * We propose a simple yet effective keyframe sampling strategy (Figure 7, Section 4.2) that significantly reduces inference time while maintaining performance for video foundation models.
> * **Encoding Context**: Many indoor scene understanding and reasoning tasks [34, 35, 40, 54] use an approach where models first encode the input, and then perform inference using lightweight decoder heads. In this context, longer encoding time does not significantly impact overall system performance.

---

> > ### Comment · Area_Chair_2Gix · 2024-08-08
> >
> > Dear reviewer,
> > The author-reviewer interaction period has started. Please read the responses provided by the authors, respond to them early on in the discussion, and discuss points of disagreement.
> > Thanks

---

> ### Comment · Reviewer_yLXP · 2024-08-08
>
> Thanks for the clarification. I will increase my score to 5.

---

> > ### Author Response · Authors · 2024-08-08
> > **Thank you for your positive feedback!**
> >
> > We appreciate the reviewer for the positive feedback. Your constructive comments and suggestions are indeed helpful for improving the paper. Also, many thanks for raising the score.
> >
> > We will continue to improve our work and release the code. If the reviewer has any follow-up questions, we are happy to discuss!

---

### Official Review · Reviewer_4P6H · 2024-07-12

**Soundness:** 3
**Presentation:** 3
**Contribution:** 3
**Rating:** 6
**Confidence:** 5

**Summary:**

This paper examines various scene encoding methods for 3D scene understanding, encompassing image, video, and 3D  models. It explores four distinct tasks:  registration, scene reasoning, visual grounding, and segmentation. The experimental results indicate that different encoding techniques excel in different tasks, underscoring the importance of selecting appropriate encoders for enhanced understanding in 3DVL.

**Strengths:**

1. This paper provides thorough analysis on different 2D and 3D performance on scene understanding. Currently, we have not seen these features compared in one framework
2. This probing framework and the insights from the experimental results are crucial for the 3D vision-language community.

**Weaknesses:**

1. Details of using 3D feature field for these tasks should be discussed. What is the baseline model for 3D grounding, QA and registration. These information should be elaborated in appendix.
2. The combination of 3D and 2D feature should be studied in different tassk.

**Questions:**

1. In 2D visual grounding, results using image feature and swin3D features are worse than M3DRef, which model do you use to conduct visual grounding?
2. See weakness above, details of these experiments should be included to make these results more convincing.

**Limitations:**

More 3D encoders should be considered, like point transformer, pointnet++, and sparse conv UNet.

---

> ### Author Rebuttal · Authors · 2024-08-07
>
> We appreciate your comments, and address your concerns as follows:
> ***
> 1. *Q: Details of using 3D feature fields for these tasks should be discussed. What is the baseline model for 3D grounding, QA, and registration.*
>
> * **3D QA**: We use 3D-LLM [35] as our baseline and backbone. We replace the original visual embedding with outputs from different VFMs, while retaining the same visual projector and large language model for answer decoding.
> * **3D grounding**: We employ Multi3DRefer [103] as our backbone. We substitute the original visual embedding with outputs from different VFMs and use the same attention-based decoder head for object-text matching.
> * **Segmentation**: Given per-point features from our unified architecture, we directly use a lightweight linear probing decoder to output semantic labels for each point.
> * **Registration**: We use REGTR [94] as our backbone, adopting its transformer encoder followed by a lightweight decoder to determine the corresponding positions of points between point clouds. However, the original REGTR evaluation with point clouds is not compatible with 2D foundation models. Hence, we modified the problem setting and created our own evaluation dataset, as detailed in Lines 277-285.
>
> For all models, we adjust the dimensions of projection and decoding layers to match the output embedding channels of the VFMs. We thank the reviewer for the suggestion, and will clarify these points in our revised manuscript.
> ***
>
> 2. *Q: The combination of 3D and 2D features should be studied in different tasks.*
>
> In Section 4.2 Line 320 and Figure 8, we studied the combination of two 2D VFMs and one 3D VFM on the semantic segmentation task, and demonstrated that combining 3D and 2D features leads to great improvement in this visual perception task.
>
> Here, as requested, we conduct the combination of 2D and 3D VFMs in registration, grounding, and VQA tasks. We select three models, including 2D, video, and 3D models, and leverage the same setting from Section 4.2.
>
> | 3DQA | ScanQA (CIDEr) ↑ | SQA3D (CIDEr) ↑
> | - | - | -
> | CLIP | 70.3 | 124.5
> | CLIP+Swin3D | 71.4 | 127.2
> | CLIP+SVD | 71.8 | 128.5
> | CLIP+SVD+Swin3D | **73.3** | **129.9**
> |
>
> | Grounding | Overall (accuracy) ↑
> | - | -
> | V-JEPA | 52.9
> | V-JEPA+Swin3D | 54.1
> | V-JEPA+DINOv2 |53.5
> | V-JEPA+DINOv2+Swin3D | **54.9**
> |
>
> | Registration | RRE (°) ↓ | RTE (m) ↓
> | - | - | -
> | SVD | 0.83 | 0.060
> | SVD+DINOv2 | 0.79 | 0.055
> | SVD+Swin3D | 0.73 | 0.053
> | SVD+DINOv2+Swin3D | **0.71** | **0.050**
> |
>
> Results show that the combination of 2D image, video, and 3D features consistently improved performance across all tasks. These results reinforce our earlier findings and lead to an important observation: the mixture of vision experts, especially those with different modalities, is a simple yet effective method for improving performance across various 3D vision tasks. It also suggests the potential for future research in developing fusion techniques for these complementary features.
> ***
>
> 3.  *Q: In 2D visual grounding, results using image features and swin3D features are worse than M3DRef, which model do you use to conduct grounding?*
>
> Thank you for your question. We'd like to clarify several points regarding our visual grounding methodology and results:
> * **Backbone Model**: As we elaborated in the answer to Q1, we use M3DRef [103] as our backbone visual grounding method, replacing only the original "vision detection module" with various visual foundation models (VFMs).
> * **Comparability**: Note that the numbers achieved by our probing model are not directly comparable to the numbers achieved by M3DRef. The reasons are:
>     * Multiple visual encoders: M3DRef utilizes both 3D object features and 2D image features during its feature extraction, effectively employing an internal mixture-of-expert mechanism to enhance their visual features and achieve better performance. In contrast, our method uses only one feature map from a single VFM in each experiment.
>     * Model finetuning: M3DRef finetunes its visual encoders on the training dataset, while we keep the visual encoders fixed and only probe their feature embeddings to clearly demonstrate different VFMs’ generic capabilities.
> * While not directly comparable, we include M3DRef's original results as a reference point, to give readers a general idea of how well various VFMs perform in a zero-shot setting relative to an established benchmark.
> We will clarify these points in our revised manuscript to prevent any misinterpretation of the results.
> ***
>
> 4. *Q: More 3D encoders should be considered, like point transformer, pointnet++, and sparse conv UNet.*
>
> We appreciate the reviewer's suggestion to consider additional 3D encoders. The encoders mentioned by the reviewer were carefully considered and not included in our submission due to the following reasons:
> * Point Transformer and PointNet++:
>     * These encoders are not typically considered visual foundation models as they lack official, generalized pretrained checkpoints.
>     * Using ScanNet-specific checkpoints (either trained by ourselves or from third parties) could introduce severe data leakage and lead to unfair comparisons with other visual foundation models.
> * Sparse Conv Unet:
>     * This architecture, exemplified by MinkowskiNet, is already utilized as the backbone for Swin3D [92], which is included in our study.
>     * We believe that the inclusion of the more advanced and scalable Swin3D effectively covers the capabilities of Sparse Conv UNet.
> * **Uni3D**: However, in the general response, we include Uni3D, a **large-scale pretrained object-centric 3D foundation model**. Please refer to the general response for the observations and analysis.
>
> We will include a discussion of these considerations in our revised manuscript to provide clarity on our methodology and model selection criteria.

---

> > ### Comment · Area_Chair_2Gix · 2024-08-08
> >
> > Dear reviewer,
> > The author-reviewer interaction period has started. Please read the responses provided by the authors, respond to them early on in the discussion, and discuss points of disagreement.
> > Thanks

---

> > ### Comment · Reviewer_4P6H · 2024-08-14
> > **Thank the authors for the rebuttal**
> >
> > I have read the rebuttal and other reviews. Most of my concerns have been solved, so I will maintain my original score as 6: Weak Accept.

---

> > > ### Author Response · Authors · 2024-08-14
> > > **Thank you for your positive feedback!**
> > >
> > > We appreciate the reviewer for the positive feedback. Your constructive comments and suggestions are indeed helpful for improving the paper.
> > >
> > > We will continue to improve our work and release the code. If the reviewer has any follow-up questions, we are happy to discuss!

---

### Official Review · Reviewer_q2Jg · 2024-07-30

**Soundness:** 3
**Presentation:** 3
**Contribution:** 3
**Rating:** 8
**Confidence:** 4

**Summary:**

This paper conducts a large-scale study to answer the unexplored question: which method (among image-based, video-based, and 3D foundation models) performs the best in 3D scene understanding? The results show that DINOv2 demonstrates superior performance, video models excel in object-level tasks, diffusion models benefit geometric tasks, and language-pretrained models show unexpected limitations in language-related tasks.

**Strengths:**

The paper is well-structured overall. The investigated question is very interesting to me. I also like the extensive experiments involved in this paper.

**Weaknesses:**

(1). What about more advanced object-centric encoders like Segment Anything (SAM) for complex 3D scene understanding? Are they better or worse than LSeg?
(2). Will the results be different when a different probing method is used (e.g., a pyramid network to aggregate multi-scale features from the foundation model)?

**Questions:**

Please see 'Weaknesses'.

**Limitations:**

The authors have discussed the limitations of this paper, and this paper has no direct negative societal impact.

---

> ### Author Rebuttal · Authors · 2024-08-07
>
> We appreciate your comments, and address your concerns as follows:
> ***
> 1. *Q: How about Segment Anything (SAM)? Does it perform better than LSeg?*
>
> Thank you for the suggestion. Here we include the performance of SAM as a visual foundation model for our evaluation benchmarks. We use the official pretrained model checkpoint with ViT-L as the backbone encoder, matching the model size with other visual foundation models in our submission.
>
> | Model | 3D VQA (CIDEr) ↑ | 3D Grounding (Accuracy) ↑ | Segmentation (mIoU) ↑ | Registration (RTE) ↓
> | - | - | - | - | -
> | LSeg | 71.0 | 50.4 | 47.5 | 0.59
> | SAM | 68.6 | 50.1 | 30.9 | 0.09
> |
>
> With the results shown above, we offer the following analysis:
> * First, it is crucial to highlight the fundamental differences between LSeg and SAM. LSeg is designed to conduct language-driven **semantic** image segmentation, providing semantic-aware representations. In contrast, SAM is primarily an **instance** segmentation model that focuses on local representations and excels in detecting edges, as illustrated in **Figure 1** of our PDF rebuttal. These distinctions result in varied performances across the four tasks in our evaluation.
> * Among the four tasks, *3D VQA* and *Semantic segmentation* require a deep semantic understanding of the 3D scenes, where LSeg naturally outperforms SAM. For *3D Grounding*, both semantic and spatial understanding are necessary; hence, LSeg and SAM exhibit similar, yet suboptimal, performance in this task. The *Registration* task, however, demands matching point clouds using distinguishable local features. Here, SAM's ability to provide precise local features positions it as a strong performer in this geometry-oriented task.
> * Overall, SAM is not well-suited for numerous downstream tasks, particularly those requiring semantic comprehension. This conclusion is consistent with previous studies, such as [A, B]. However, we additionally reveal that it excels in tasks benefiting from robust local feature representation.
>     * *[A] AM-RADIO: Agglomerative Vision Foundation Model - Reduce All Domains Into One, CVPR 2024*
>     * *[B] SAM-CLIP: Merging Vision Foundation Models towards Semantic and Spatial Understanding, CVPRW 2024*
>
> ***
> 2. *Q: Will the results be different when a different probing method is used (e.g., a pyramid network to aggregate multi-scale features from the foundation model)?*
>
> We have conducted additional experiments to evaluate the impact of multi-scale feature aggregation on various visual foundation models (VFMs). Our analysis focused on CLIP, SAM, StableDiffusion, and Swin3D, which represent a diverse range of VFMs with different pretraining settings and input modalities.
>
> For CLIP and SAM which use ViTs as backbones, we follow [6,97] and split the layers into four equally sized blocks and extract features after the last three blocks (i.e., 12-th, 18-th, and 24-th layers in ViT-L). For StableDiffusion and Swin3D, the decoding portion of the UNet and MinkowskiNet consists of feature upsampling blocks. We extract features after three of these upsampling blocks.
>
> | Model | Feature Configuration | 3D VQA (CIDEr) ↑ | 3D Grounding (Accuracy) ↑ | Segmentation (mIoU) ↑ | Registration (RTE) ↓
> | - | - | - | - | - | -
> | CLIP | Single-scale | 70.3 | 50.4 | 3.4 | 0.44
> | | Multi-scale Aggregation | 70.9 | 51.1 | 3.8 | 0.28
> | SAM | Single-scale | 68.6 | 50.1 | 30.9 | 0.09
> | | Multi-scale Aggregation | 69.0 | 50.5 | 31.7| 0.09
> | StableDiffusion | Single-scale | 68.2 | 50.6 | 42.6 | 0.09
> | | Multi-scale Aggregation | 69.8 | 51.7 | 29.8 | 0.07
> | Swin3D | Single-scale | 62.3 | 43.6 | 18.1 | 0.71
> | | Multi-scale Aggregation | 70.0 | 51.6 | 35.2 | 0.23
> |
>
> From the results we observe:
> * **Performance improvement**: Most models and tasks showed improved performance with multi-scale feature aggregation. This improvement was more pronounced in CNN-based architectures compared to Transformer-based methods.
> * **Architectural differences**: CNN-based (or UNet-based) models benefited more due to the complementary nature of features from different convolutional layers with varying receptive fields. In contrast, Transformer-based (or ViT-based) models, with their fully connected and attended layers, showed less significant improvements.
> * **Model-specific observations**: StableDiffusion exhibited a performance drop after multi-scale aggregation for the semantic segmentation task. Further analysis revealed that its final layers focus on high-frequency textures, which are unsuitable for tasks like segmentation (yielding only 16.2 mIoU when used alone). These results provide valuable insights for the mixture-of-experts approach, suggesting the importance of carefully selecting which layers or models to include in the expert pool to achieve optimal performance.
>
> We will add these results and discussion into the analysis section of our revised manuscript.

---

> > ### Comment · Area_Chair_2Gix · 2024-08-08
> >
> > Dear reviewer,
> > The author-reviewer interaction period has started. Please read the responses provided by the authors, respond to them early on in the discussion, and discuss points of disagreement.
> > Thanks

---

> > ### Comment · Reviewer_q2Jg · 2024-08-13
> >
> > Thank you for the rebuttal. My concerns are addressed by the authors. Therefore, I will keep my score.

---

> > > ### Author Response · Authors · 2024-08-13
> > > **Thank you for your positive feedback!**
> > >
> > > We appreciate the reviewer for the positive feedback. Your constructive comments and suggestions are indeed helpful for improving the paper.
> > >
> > > We will continue to improve our work and release the code. If the reviewer has any follow-up questions, we are happy to discuss!

---

### Author Rebuttal · Authors · 2024-08-07

# General Response

***
We are thankful for the feedback and suggestions from all the reviewers. We are glad that the reviewers recognize our intriguing and meaningful insights for the entire 3D vision and multi-modal community (4P6H, vFsT), representative tasks and wholesome coverage of visual foundation models (yLXP, vFsT), extensive and comprehensive experiments and analysis (q2Jg, 4P6H, vFsT), and well-structured manuscript (q2Jg, 4P6H, yLXP).

We address each of the reviewers’ concerns in the individual response. Here, we would like to highlight the **key objectives and contributions** of our paper:
* Being the first comprehensive study on the role of image, video, and 3D visual foundation models (VFMs) in 3D multi-modal perception and reasoning scenarios, our work focuses on thoroughly understanding the strengths and limitations of different VFMs. Instead of optimizing the performance for a single or a few tasks, we primarily promote the breadth and generalizability of our discovery. We achieve this by employing the most straightforward, simplest, and unified design across a wide range of tasks, including 3D question answering, object grounding, semantic segmentation, and geometric registration.
* In addition to our key observations and insights as demonstrated in our original manuscript (Lines 50-61), we also summarize several universal and generalizable principles: (1) Empirically, no single VFM can uniformly dominate all visual tasks. However, leveraging 2D VFMs in 3D scene perception and multi-modal reasoning tasks consistently yields significant improvements. (2) Pretraining tasks and input modality have significant influence over a foundation model’s strengths and weaknesses. However, the straightforward yet efficient mixture of vision-experts boosts performance for all tasks by effectively leveraging complementary knowledge from different VFMs.

Therefore, we kindly suggest that our contributions lie in presenting the applicability of our discovery and the intriguing emergent behaviors across a wide range of tasks, instead of focusing on scaling up and optimizing individual tasks. We are grateful that reviewers (4P6H,vFsT) approach our paper from this perspective.

**Additional Experiments**: In addressing the reviewers’ main concerns, we provide additional experiments in the individual responses. A summary of new results is shown below:
* **Mixture-of-Vision-Experts (MoVE)**: We validate that combining multi-layer features from the same visual model and features from multiple visual models both lead to a consistent boost of performance across different tasks. This reinforces our earlier findings about the significance of leveraging Mixture-of-Vision-Experts (MoVE) in 3D scene understanding and multi-modal reasoning scenarios. Please refer to the response to reviewers q2Jg and 4P6H for more details.
* **SAM**: We provide the evaluation of the Segment Anything Model and its comparison with LSeg. Due to the max word restriction for a response, please refer to the separate response to reviewer q2Jg for more details.
* **Uni3D**: We provide the evaluation of a latest large-scale object-centric pretrained model Uni3D.

    **Implementation**: Following the part segmentation details in Uni3D's appendix (Sec. B), we used Uni3D-giant, selecting features from the 16th, 28th, and 40th (last) layers to form grouped point patches. We then employed PointNet++'s feature propagation to upsample group features into point-wise features. It's worth noting that Uni3D's ScanNet visualizations in their paper were achieved by applying Uni3D to each object instance based on ground truth instance segmentation, not by direct application to the whole scene.

    The results are shown in the following Table.

| Model | 3D VQA (CIDEr) ↑ | 3D Grounding (Accuracy) ↑ | Segmentation (mIoU) ↑ | Registration (RTE) ↓
| - | - | - | - | -
| Swin3D | **70.9** | **51.6** | **35.2** | 0.23
| Uni3D | 63.1 | 51.1 | 2.7 | **0.08**
|

* **Uni3D Results and Observations**:
    * Scene-level tasks (3D VQA and Semantic Segmentation): Uni3D underperforms compared to the scene-level pretrained Swin3D model. This is likely due to the object-centric pretraining recipe of Uni3D, causing the failure of feature extraction on large scenes with orders of magnitude more points than single objects.
    * Object-centric tasks (3D object grounding): Uni3D achieves comparable results with Swin3D. However, some grounding questions require not only object-level semantics, but also inter-object relationship and global room information, which Uni3D lacks. We believe combining object-centric and scene-level representations would be a great future direction to achieve better object grounding in a complex 3D scene.
    * Registration: Uni3D achieves better performance than Swin3D, suggesting that geometric knowledge from object-centric pretraining generalizes well to scene-level geometric matching, especially given the task's use of downsampled partial scenes bridging the distribution gap between object-level and scene-level point clouds.

---

### Decision · Program_Chairs · 2024-09-25

**Decision:**

Accept (poster)

**Comment:**

The paper presents a comprehensive study that probes various visual encoding models for 3D scene understanding, identifying the strengths and limitations of each model across different scenarios.

All reviewers recommend accepting the paper. The experiments are thorough, and the insights are compelling. Please ensure the reviewers' comments are incorporated into the final version.